# Prognosis of Atrial Fibrillation with or without Comorbidities: Analysis of Younger Adults from a Nationwide Database

**DOI:** 10.3390/jcm11071981

**Published:** 2022-04-01

**Authors:** Valentin Mertz, Yves Cottin, Sid Ahmed Bentounes, Julie Pastier-Debeaumarché, Romain Didier, Julien Herbert, Marianne Zeller, Gregory Y. H. Lip, Laurent Fauchier

**Affiliations:** 1Service de Cardiologie, CHU Dijon Bourgogne, 21000 Dijon, France; valentin.mertz@chu-dijon.fr (V.M.); julie.debeaumarche@chu-dijon.fr (J.P.-D.); romain.didier@chu-dijon.fr (R.D.); 2Department of Cardiology, CHU Dijon Bourgogne, 21000 Dijon, France; 3Service de Cardiologie, CHU Trousseau et Université François Rabelais, 37000 Tours, France; sidahmed_bentounes@yahoo.fr (S.A.B.); j.herbert@chu-tours.fr (J.H.); laurent.fauchier@univ-tours.fr (L.F.); 4Service D’information MéDicale, D’épidémiologie et D’économie de la Santé, Faculté de Médecine, Centre Hospitalier Universitaire, Université de Tours, 37000 Tours, France; 5Physiopathologie et Epidémiologie Cérébro-Cardiovasculaires, University of Bourgogne Franche Comté, 21000 Dijon, France; marianne.zeller@u-bourgogne.fr; 6Liverpool Centre for Cardiovascular Science, University of Liverpool, Liverpool Heart and Chest Hospital, Liverpool L14 3PE, UK; gregory.lip@liverpool.ac.uk

**Keywords:** atrial fibrillation, stroke, mortality, heart failure, heart disease, comorbidities, prognosis

## Abstract

Objective: To assess the prognosis of AF patients with or without cardiac or extra-cardiac concomitant conditions. Participants and Methods: All consecutive patients diagnosed with AF admitted to French hospitals between 2011 and 2020 were identified. Patients were classified into four groups: (1) > 60 yo; (2) with known cardiac disease (KCD group); (3) with extra-cardiac comorbidities (ECC); and 4) AF without KCD or ECC (“Lone AF”). Results: Altogether 2,435,541 patients were identified, from which 2,203,702 patients aged >60 years and 231,839 patients aged <60 years (with KCD (55.2%), with ECC (14.7%) and with “Lone AF” (30.1%)). During follow-up, the incidences of all-cause and CV deaths were 13.7%, 5.7%, 6.2%, and 2.3%, and 4.2%, 1.7%, 0.8%, and 0.3% in the older than 60 yo group, KCD group, ECC group and “Lone AF” AF group, respectively. In the age and sex-adjusted analysis (patients < 60 yo), patients with AF and KCD had worse outcomes than patients with “Lone AF” for all major cardiac events. Conclusion: There are three distinct prognostic criteria based on the presence or lack of HD or extra-cardiac concomitant comorbidities. Patients in the so-called “Lone AF” group remain severe in terms of CV events but still with a lower incidence than the patients with associated KCD or ECC. The presence of KCD or ECC makes it possible to distinguish a profile in terms of events that are very different between the patients.

## 1. Introduction

Clinical AF results from an interaction between triggers and sustaining mechanisms (“substrates”) composed of electrical and/or structural components [1,2]. A wide number of conditions are associated with atrial fibrillation (AF), usually classified into two groups: (a) heart disease (e.g., coronary heart disease, heart failure, valvular heart disease, hypertension); or (b) extra-cardiovascular (CV) concomitant conditions (e.g., diabetes mellitus, pulmonary disease, sleep apnea syndrome, etc.) [3,4]. Current ESC guidelines indicate that the term “Lone AF” is potentially confusing and should be abandoned because a cause may be present in every patient [4]. It may, however, still be used in daily practice considering its simplicity and the lack of a better term. “Lone AF” may be defined as AF in younger adults (age < 60 years) and with a lack of obvious associated CV or extra CV conditions. Moreover, most studies about “Lone AF” did not exclude older patients even if it is established that 60 years is arbitrary and not based on validated pathophysiological justification [3].

In addition, studies on the prognosis of “Lone AF” are inconsistent, likely due to the result of the heterogeneity in definitions, comorbidities, study population, and duration of follow-up [3,4,5]. The prognosis of AF is primarily, but not always, determined by cardiac comorbidities and small cohort studies have suggested a better prognosis in AF patients without known cardiac disease (KCD) [5,6].

The aim of the present study was to assess on a national scale: a) the prognosis of patients hospitalized with AF with and without known cardiac disease (KCD) or extra-cardiac comorbidities (ECC), and b) to determine the prognostic value of these conditions associated with AF.

## 2. Methods

### 2.1. Study Design

This retrospective cohort study was based on data from the French administrative hospital discharge database *(Programme de Médicalisation des Systèmes d’Information—PMSI*), which collects collected medical information according to the International Classification of Diseases, Tenth Revision (ICD-10) and medical procedures recorded according to the national nomenclature Classification Commune des Actes Medicaux (CCAM). The reliability of PMSI data has already been assessed and this database has previously been used to study patients with CV conditions, including AF and KCD [7,8,9]. The French Data Protection Authority granted access to the PMSI data, and procedures for data collection and management were approved by the independent national ethics committee for human rights in France (*Commission Nationale de l’Informatique et des Libertés—CNIL*), which ensures that all information is kept confidential and anonymous in compliance with the Declaration of Helsinki (*authorization number 1897139*).

### 2.2. Study Population

From January 2011 to December 2020, 2,435,541 adults (age ≥18 years) were hospitalized with a diagnosis of AF as the principal diagnosis (i.e., the condition justifying hospital admission), a related diagnosis), or a significantly associated diagnosis (i.e., comorbidity or associated complication). Starting from this database of patients with AF, we extracted patients >60 years (2,203,702 patients, 90.48%), and patients ≤ 60 years, allowing us to identify our cohort of interest of 231,839 patients with a diagnosis of AF (9.52%) in Figure 1. We identified 4 groups of patients: (1) patients > 60 yo, (2) group of AF patients with known cardiac disease ≤ 60 yo (AF-KCD) with 127,954 patients (55.2%); (3) group of AF patients with extra-cardiac comorbidities ≤60 yo (AF-ECC) with 34,011 patients (14.7%); (4) group with so-called “Lone AF” ≤60 yo with 69,874 patients (30.1%) defined as those not included in group 2 or group 3. Extra-cardiac comorbidities included: diabetes mellitus, obesity, vascular disease, alcohol-related diagnoses, abnormal renal function, lung diseases, sleep apnea syndrome, chronic obstructive pulmonary disease, thyroid diseases, and inflammatory diseases. 

For each hospital stay, we evaluated the CHA_2_DS_2_VASc score and a proxy of the HAS-BLED score (i.e., all items except labile INR, which was unavailable, as in most administrative databases using ICD codes). We performed an analysis with adjustment on age and sex with a hazard ratio associated with the type of AF (with “Lone AF” as reference) for the incident outcome of interest.

For the supplemental analysis, we identified 6 subgroups: (1) Patients ≤ 60 yo with KCD; (2) Patients ≤ 60 yo with ECC; (3) Patients ≤ 60 yo with “lone AF”; (4) Patients > 60 yo with KCD; (5) Patients > 60 yo with ECC, (6) Patients > 60 yo without KCD or ECC (Appendix A).

### 2.3. Outcomes

We evaluated the incidence of all-cause death, CV death, non-CV death, ischemic stroke, and rehospitalization for HF (identified as the main diagnosis for hospital stay). The endpoints were evaluated from January 2010 to follow-up and the starting date of inclusion was that of the first diagnosis of AF. Information on outcomes during follow-up was obtained by analyzing the PMSI codes for each patient. The cause of death was identified based on the main diagnosis during the hospital stay that resulted in death. 

### 2.4. Statistical Analysis 

Qualitative variables are described as frequencies and percentages and quantitative variables as means (standard deviations (SD)) and medians (interquartile ranges) for data that were not normally distributed. Comparisons were made using Chi-square tests for categorical variables and the Student’s t-test for continuous variables. For the outcomes analysis, the incidence rates (%/year) for each outcome of interest during follow-up were estimated in the subgroups of interest. Multivariable analysis for clinical outcomes was performed using a Cox model to calculate the hazard ratio (HR) and 95% confidence interval (CI) for comparing subgroups of interest with adjustment on age at baseline and sex. A two-sided *p*-value of <0.05 was considered statistically significant. All analyses were performed using Enterprise Guide 7.1 (SAS Institute Inc., SAS Campus Drive, Cary, NC, USA) and STATA version 16.0 (Stata Corp, College Station, TX, USA).

## 3. Results

### 3.1. Baseline Characteristics of Patients Aged under 60 Years

Overall, we identified 231,839 patients under 60 years old in the database between January 2011 and December 2020 including 69,874 patients (30.1%) in the group “Lone AF”, 127,954 patients (55.2%) in the group AF-KCD, and 34,011 patients (14.7%) in the group AF-ECC (Figure 1, Table 1 and Appendix A). Patients with “Lone AF” were younger than patients with KCD or ECC, respectively, 49.1 ± 9.9 and 53.5 ± 6.8 or 51.1 ± 8.6 yo (*p* < 0.0001). As expected, patients with “Lone AF” had lower CHA_2_DS_2_VASc score than patients with KCD or ECC, respectively, 0.3 ± 0.7 and 1.9 ± 1.1 or 0.6 ± 0.6 (*p* < 0.0001).

### 3.2. Clinical Outcomes in Patients with AF of Patients Aged under 60 Years

Table 2 summarizes the clinical outcomes during the whole follow-up (mean (SD) 2.3 (2.5), median (IQR) 1.3 (0.1–4.1) years) in AF patients under 60 years old. The incidence rate of all-cause death was higher in the AF-KCD and AF-ECC groups than in the “Lone AF” group: 5.7% 6.2% and 2.3% per year, respectively. The incidence of CV death was higher in the AF patients with KCD than in the ECC and “Lone AF” groups: 1.7% vs. 0.8% and 0.3% per year, respectively. The incidence of non-CV death was higher in the AF patients with ECC than in the KCD and “Lone AF” groups: 5.4% vs. 4.0% and 2.0% per year, respectively (Table 2 and Figure 2). The rates of ischemic stroke and rehospitalization for HF were higher in the AF patients with KCD group: 1.3% per year and 5.6% per year (Table 2, Figure 3).

### 3.3. Clinical Outcomes in the Cohort Aged under 60 Years

During follow-up in the analysis adjusted on age and sex, the all-cause death patients with AF-KCD or AF-ECC had a higher risk than patients with “Lone AF”, respectively, for the adjusted hazard ratio (HR (95%CI)), 2.13 (2.05–2.21) or 2.46 (2.36–2.57) (*p* < 0.0001) (Table 3, Figure 2).

Regarding CV death, patients with AF-ECC or AF-KCD had a higher risk than patients with “Lone AF”, HR (95%CI), 2.39 (2.12–2.69), (*p*< 0.0001) or 4.80 (4.37–5.27), (*p* < 0.0001) (Table 3, Figure 2). Nevertheless, we observed that patients with AF-ECC had less risk of CV death than patients with AF-KCD, HR (95%CI), 0.50 (0.46–0.54), (*p* < 0.0001) (Table 3, Figure 2).

For non-CV death, patients with AF-ECC and AF-KCD had a higher risk than patients with “lone AF”, respectively, HR (95%CI), 2.47 (2.35–2.59) (*p* < 0.0001) and 1.72 (1.65–1.79) (*p* < 0.0001) (Table 3, Figure 2). We observed that patients with AF-ECC had a worse prognosis for non-CV death than AF-KCD, HR (95%CI), 1.44 (1.39–1.49), (*p* < 0.0001) (Table 3, Figure 2).

The outcome for ischemic stroke patients with AF-ECC and AF-KCD was a higher risk than patients with “Lone AF” (Table 3, Figure 2). Patients with AF-ECC had a lower risk of ischemic stroke than patients with AF-KCD, HR (95%), 0.69 (0.64–0.75), (*p* < 0.0001) (Table 3, Figure 3).

For the outcome of rehospitalization for heart failure (HF), patients with AF-ECC and AF-KCD had higher risks than those with “Lone AF”, adjusted HRs (95%CI), 1.67 (1.55–1.81) (*p* < 0.0001) and 5.76 (5.44–6.09) (*p* < 0.0001), respectively (Table 3, Figure 3). Patients with AF-ECC had less risk of rehospitalization for heart failure than patients with AF and KCD, HR (95%CI), 0.29 (0.27–0.31), (*p* < 0.0001) (Table 3, Figure 3).

### 3.4. Clinical Outcomes of Patients Aged below or above 60 Years 

Patients over 60 years old had markedly higher incidences of clinical events that were on average twice higher than those in patients under 60 years old with KCD (for example, 2.5% per year vs. 1.3% per year for ischemic strokes) (Table 2).

For all outcomes, there was a higher risk of clinical events in patients above 60 years old compared to the three groups of patients under 60 years old (“Lone AF”, AF-KCD, and AF-ECC) (Table 3). Appendix A shows the flow chart of the two groups under and over 60 years and the distribution of the three groups (“Lone AF”, AF-KCD, and AF-ECC). Older patients were more likely to have AF associated with KCD and less likely to have AF associated with neither KCD nor ECC. Appendix A presents the baseline characteristics of adult patients hospitalized in France (2011–2020) with a history of AF (age ≤60 and age >60) and Appendix A present the cumulative incidences of major events according to age (≤60 or age >60). Beyond older age, patients aged >60 were less predominantly male and had a higher prevalence for most comorbidities. Beyond the higher risk for all clinical events in patients >60 years old, the higher risk of all-cause death with AF-ECC in patients aged ≤60 was not found in those aged >60 where patients with AF-KCD had the highest mortality. Regarding ischemic stroke, there were more clear differences in incidences in the three groups of patients (highest incidence of ischemic stroke in patients with AF-KCD and the lowest incidence in patients with “Lone AF”) in patients aged ≤60 than in those aged >60.

## 4. Discussion

In this large nationwide study, of patients aged under 60 years with AF, three distinct clinical entities were distinguished based on the association of AF: (a) with known cardiac disease (KCD); (b) with extra-cardiac comorbidities (ECC); or (c) so-called “Lone AF”. “Lone AF” represented 2.8% of the initial cohort of all AF patients but accounted for 30% of AF in patients under 60 years of age.

Our principal findings are as follows: (i) most patients hospitalized had AF-KCD (55.2%); (ii) in the adjusted analysis, patients with AF-KCD or AF-ECCC had a higher incidence of all-cause death and CV death than those with “Lone AF”; and (iii) patients in the “KCD AF” group had the worse prognosis for CV events (CV death, ischemic stroke, or rehospitalization for HF) than the patients with associated KCD. Our work suggests a simple approach to prognostic risk stratification: 1. age < 60 years old, 2. presence of heart disease, and 3. presence of comorbidities; it provides major information in terms of prognosis. 

### 4.1. Comorbidities in Clinical Trials of AF

In 1954, Evans and Swann used the term “Lone AF” to describe patients for whom “subsequent investigation shows that heart disease is absent” [9]. Over the past 30 years, there has been an increase in knowledge of AF, and the number of conditions known to be associated with AF has increased, such as inflammation, or obstructive sleep apnea syndrome [3,9,10,11,12,13,14]. The number of conventional risk factors increases with the aging of the population, and the emerging risk factors also increase sharply [3]. 

Our study arbitrarily categorized the patients into three groups, although there is actually no formal validated stratification concerning the comorbidities associated with AF and its global prognosis. Many studies have studied the percentage of conditions associated with AF. The recent work by Khan et al. compiled 134 randomized studies and reported that 4.5% of AF patients had obstructive sleep apnea, 35.8% had coronary artery disease (CAD), 66.4% had diabetes mellitus, 75.4% had hypertension, 12.7% had dyslipidemia, 9.7% had chronic kidney disease (CKD), 44% had heart failure (HF), 45.5% had cerebrovascular accidents, and 9% had the chronic obstructive pulmonary disease (COPD) [13]. Indeed, multimorbidity is common in patients with AF and both non-cardiac and cardiac comorbidities clusters may be associated with an increased risk of major adverse outcomes of different manners [14,15,16,17,18,19,20,21,22,23,24,25,26,27,28].

The RE-LY AF registry, which enrolled 15,400 patients in 47 countries, compared the data according to the presence or absence of traditional risk factors [14]. Traditional risk factors were defined by age > 60 years, hypertension, coronary artery disease, heart failure, left ventricular hypertrophy, congenital heart disease, pulmonary disease, valve heart disease, hyperthyroidism, and prior cardiac surgery [14]. In this registry, only 796 patients (5%) had no traditionally defined risk factors. Our results are different, with 30.1% of “Lone AF” after exclusion of patients over 60 years of age who represented 90% of the initial cohort (Figure 1). This point is valid as the average age of the 796 patients of the RELY registry cohort was 45.7 ± 10.1 years old. In addition, after matching on age and region (1:3), AF patients without traditional risk factors had a 1-year lower stroke and heart failure hospitalizations occurrence [14].

All these data confirm the multi-complexity of the AF population, but our results underline the interest of stratification to optimize the understanding of AF and also to possibly optimize the management, whether pharmacological or non-pharmacological, of AF. Indeed, recent guidelines have recommended a more holistic or integrated approach to AF care, including stroke prevention, patient-centered decisions on rate- or rhythm control, and the optimization of comorbidities [4,5,6,7,8,9,10,11,12,13,14,15,16,17,18,19,20,21,22,23,24,25,26,27,28,29]. Adherence to such an approach has been associated with improved clinical outcomes [30,31]. 

### 4.2. Known Heart Disease in Young Patients (≤60 Years Old) with AF

It is generally believed that many young patients have AF without comorbidities. The prevalence of AF without comorbidities varies between 1% and 68%, depending on the age of AF onset, type of AF, and “Lone AF” definition [3,15,16]. Our analysis focused on patients under 60 years of age with AF, but our data are in agreement with De With et al., who included 498 patients under the age of 60 with 9% of heart failure, 10% of coronary artery disease, 4% of COPD, 6% of stroke [16]. Like others studies of the literature, our data show that amongst young patients aged <60 years, only 30.9% had AF without comorbidities, which may still be an overestimation since this definition did not include subclinical vascular disease.

Of note, we did not have data on familial AF, which may be present with incidences ranging from 5% to 46% [16,17,18]. The age of 60 years is commonly used in these studies and our work shows that the evolution of patients is clearly different with a markedly higher incidence of CV events beyond the age of 60. Our study therefore only covered 10% of the initial population but enabled us to define relatively homogeneous groups in this specific setting, providing relevant information in terms of CV events at the follow-up.

### 4.3. Cardiovascular Outcomes: Higher Incidence of Ischemic Stroke and Rehospitalization in Patients of the KCD Group Compared to the ECC Group

AF in the absence of traditional risk factors is often considered a benign disease [19]. In our study, patients in the “Lone AF” group remain severe in terms of CV events but still with an incidence lower than the patients with associated KCD or ECC [20,21,22]. 

Moreover, studies concerning the prognosis of “lone AF” report contradictory results, likely because the multi-complexity of the AF population and the prognosis of AF is primarily, but not always, determined by its cardiac and associated comorbidities [3,23]. The originality of our work is to have separated the comorbidities into two groups: CV and non-CV comorbidities. If the percentage of patients with heart disease is greater than 50% (55.19%), the percentage of patients with AF and extra-cardiac pathology is around 15%. Although there are no data in the literature, the distribution of pathologies that are exclusively extra-cardiac is consistent [14].

The patients without known heart disease but presenting comorbidities associated with AF are, in our study, mainly patients with obesity (28.9%), excessive alcohol consumption (27.1%), lung disease (24.2%), COPD (10.7%), or having had previous cancer (15.7%). Our data are consistent with the Danish Diet, Cancer, and Health study, which found that moderate alcohol intake (about 1.5 drinks daily) increased AF risk by 25% to 46% among men [24,25]. 

On the other hand, patients in the ECC group compared to patients with KCD are associated with a significant increase in all-cause death and non-CV death but less CV death, ischemic stroke, or rehospitalization for heart failure.

In our study, patients with “AF and KCD” were older and had a higher CHA_2_DS_2_VASc score, consistent with the literature and confirming a relationship between this score and CV events. Of note, COPD is an independent and strong predictor of incident AF. The presence of COPD increases the HR for incident AF about five-fold in patients with a CHA_2_DS_2_VASc score, while the coexistence of a CHA_2_DS_2_VASc score ≥2 minimizes the prognostic significance of COPD [26]. 

## 5. Limitations 

The main limitation is inherent to the retrospective, observational nature of the study and its potential biases. The diagnoses and occurrence of outcomes in our study were based on the diagnostic codes registered by a responsible physician and were not further checked externally. However, as coding of complications is linked to reimbursement and is regularly controlled, it is expected to be of good quality. We had no information about death occurring outside the hospitals. Our large population of patients hospitalized with AF represents a heterogeneous group of patients admitted with various kinds of illnesses and severities, which may have affected prognosis. Another limitation is the lack of information on antithrombotic drug use and its possible changes during follow-up, as data regarding these therapies were not available in the complete database. There is a similar issue regarding the lack of information in terms of therapies recommended for HD, AF, and other comorbidities beyond the representative sample from our analysis. Definite conclusions for comparisons between groups may not be fully appropriate even though multivariable matching was performed as it cannot fully eradicate the possible confounding variables between these groups. The study was not racially and ethnically diverse and our findings may not be generalizable to other populations. 

## 6. Conclusions

In a national cohort of young patients hospitalized with AF, there are three distinct prognostic criteria based on the presence or lack of HD or extra-cardiac concomitant comorbidities. Patients in the so-called “Lone AF” group remain severe in terms of CV events but still with a lower incidence than the patients with associated KCD or ECC. The presence of KCD or ECC makes it possible to distinguish a profile in terms of events that are very different from other patients, in particular in terms of CV mortality from ischemic stroke or rehospitalization for heart failure. 

## Figures and Tables

**Figure 1 jcm-11-01981-f001:**
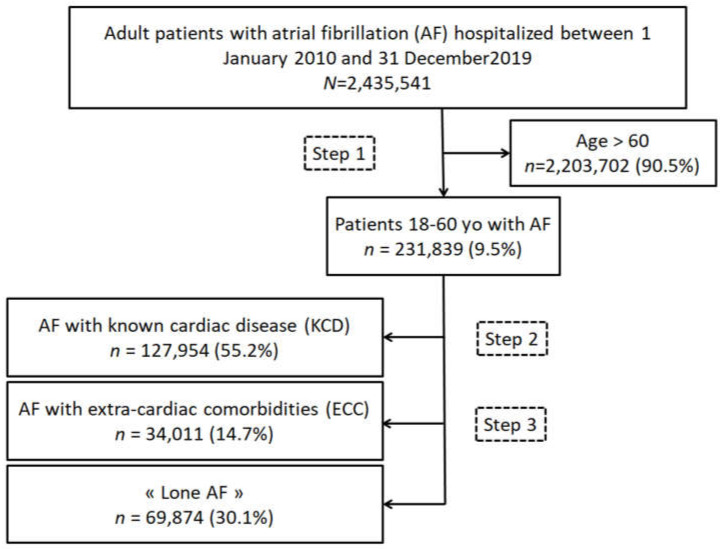
Flow chart of the study population.

**Figure 2 jcm-11-01981-f002:**
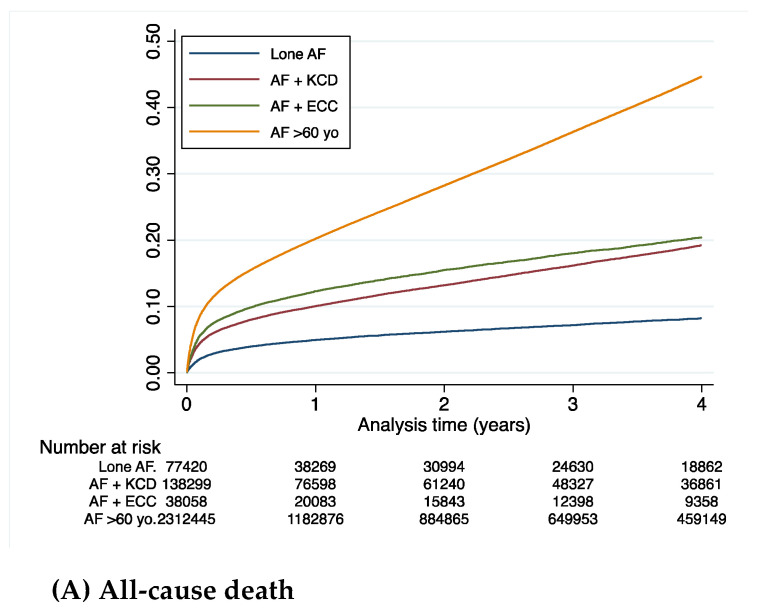
Cumulative incidences of all-cause death (**A**), cardiovascular death (**B**), or non-cardiovascular death (**C**) during follow-up in patients with AF (< 60 years old) (KCD: known cardiac disease; ECC: extra-cardiac comorbidities).

**Figure 3 jcm-11-01981-f003:**
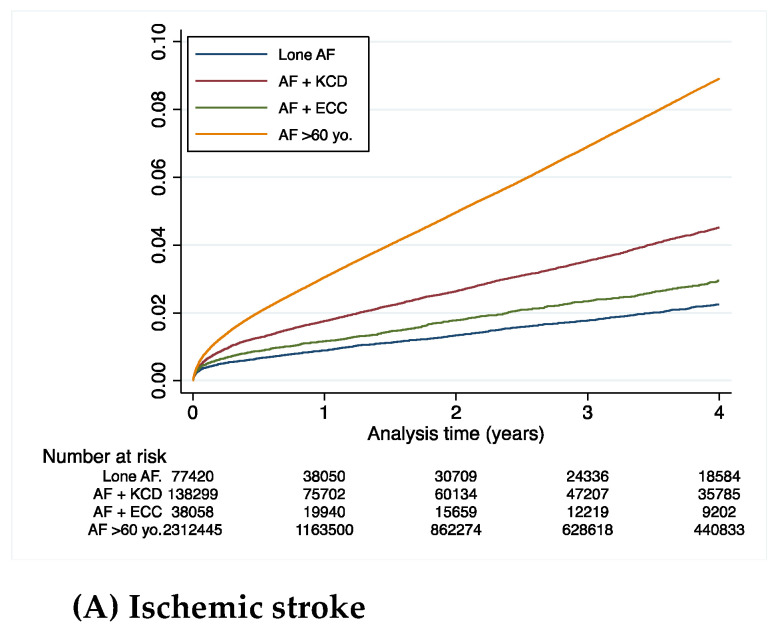
Cumulative incidences of ischemic stroke (**A**) or rehospitalization for HF (**B**) during follow-up in patients with atrial fibrillation (<60 years old) (KCD: known cardiac disease; ECC: extra-cardiac comorbidities).

**Table 1 jcm-11-01981-t001:** Baseline characteristics of patients hospitalized in France (2011–2020) aged 18–60 with history of AF (KCD: known cardiac disease; ECC: extra-cardiac comorbidities).

	Total	Lone AF	AF with KCD	*p* vs. Lone AF	AF with ECC	*p* vs. Lone AF
	(*n* = 231,839)	(*n* = 69,874)	(*n* = 127,954)		(*n* = 34,011)	
Age (years), mean ± SD	51.8 ± 8.4	49.1 ± 9.9	53.5 ± 6.8	<0.0001	51.1 ± 8.6	<0.0001
Sex (male), *n* (%)	167,893 (72.4)	50,489 (72.3)	94,014 (73.5)	<0.0001	23,390 (68.8)	<0.0001
Hypertension, *n* (%)	79,398 (34.2)	0 (0.0)	79,398 (62.1)	<0.0001	0 (0.0)	-
Diabetes mellitus, *n* (%)	33,565 (14.5)	0 (0.0)	28,114 (22.0)	<0.0001	5451 (16.0)	<0.0001
Smoker, *n* (%)	42,942 (18.5)	4672 (6.7)	30,622 (23.9)	<0.0001	7648 (22.5)	<0.0001
Dyslipidemia, *n* (%)	38,179 (16.5)	2583 (3.7)	32,518 (25.4)	<0.0001	3078 (9.1)	<0.0001
Obesity, *n* (%)	44,952 (19.4)	0 (0.0)	35,124 (27.5)	<0.0001	9828 (28.9)	<0.0001
Heart failure with congestion, *n* (%)	59,249 (25.6)	0 (0.0)	59,249 (46.3)	<0.0001	0 (0.0)	-
History of pulmonary edema, *n* (%)	8188 (3.5)	0 (0.0)	8188 (6.4)	<0.0001	0 (0.0)	-
Mitral regurgitation, *n* (%)	13,036 (5.6)	0 (0.0)	13,036 (10.2)	<0.0001	0 (0.0)	-
Mitral stenosis, *n* (%)	1463 (0.6)	0 (0.0)	1463 (1.1)	<0.0001	0 (0.0)	-
Aortic regurgitation, *n* (%)	5399 (2.3)	0 (0.0)	5399 (4.2)	<0.0001	0 (0.0)	-
Aortic stenosis, *n* (%)	6352 (2.7)	0 (0.0)	6352 (5.0)	<0.0001	0 (0.0)	-
Previous endocarditis, *n* (%)	2067 (0.9)	0 (0.0)	2067 (1.6)	<0.0001	0 (0.0)	-
Dilated cardiomyopathy, *n* (%)	23,809 (10.3)	0 (0.0)	23,809 (18.6)	<0.0001	0 (0.0)	-
Coronary artery disease, *n* (%)	42,025 (18.1)	0 (0.0)	42,025 (32.8)	<0.0001	0 (0.0)	-
Previous MI, *n* (%)	12,445 (5.4)	0 (0.0)	12,445 (9.7)	<0.0001	0 (0.0)	-
Previous PCI, *n* (%)	10,238 (4.4)	0 (0.0)	10,238 (8.0)	<0.0001	0 (0.0)	-
Previous CABG, *n* (%)	4853 (2.1)	0 (0.0)	4853 (3.8)	<0.0001	0 (0.0)	-
Vascular disease, *n* (%)	30,992 (13.4)	0 (0.0)	28,952 (22.6)	<0.0001	2040 (6.0)	<0.0001
Left BBB, *n* (%)	2820 (1.2)	204 (0.3)	2498 (2.0)	<0.0001	118 (0.3)	0.13
Right BBB, *n* (%)	3241 (1.4)	536 (0.8)	2375 (1.9)	<0.0001	330 (1.0)	0.001
Previous pacemaker or ICD, *n* (%)	5009 (2.2)	307 (0.4)	4470 (3.5)	<0.0001	232 (0.7)	<0.0001
Ischemic stroke, *n* (%)	9186 (4.0)	1657 (2.4)	6300 (4.9)	<0.0001	1229 (3.6)	<0.0001
Intracranial bleeding, *n* (%)	3178 (1.4)	480 (0.7)	2107 (1.6)	<0.0001	591 (1.7)	<0.0001
Alcohol related diagnoses, *n* (%)	28,451 (12.3)	0 (0.0)	19,240 (15.0)	<0.0001	9211 (27.1)	<0.0001
Abnormal renal function, *n* (%)	7151 (3.1)	0 (0.0)	6517 (5.1)	<0.0001	634 (1.9)	<0.0001
Lung disease, *n* (%)	28,981 (12.5)	0 (0.0)	20,735 (16.2)	<0.0001	8246 (24.2)	<0.0001
Sleep apnea syndrome, *n* (%)	16,188 (7.0)	0 (0.0)	12,551 (9.8)	<0.0001	3637 (10.7)	<0.0001
COPD, *n* (%)	15,804 (6.8)	0 (0.0)	11,750 (9.2)	<0.0001	4054 (11.9)	<0.0001
Liver disease, *n* (%)	14,342 (6.2)	613 (0.9)	10,454 (8.2)	<0.0001	3275 (9.6)	<0.0001
Thyroid diseases, *n* (%)	14,411 (6.2)	0 (0.0)	9101 (7.1)	<0.0001	5310 (15.6)	<0.0001
Inflammatory disease, *n* (%)	9179 (4.0)	0 (0.0)	6187 (4.8)	<0.0001	2992 (8.8)	<0.0001
Anemia, *n* (%)	24,523 (10.6)	2725 (3.9)	17,754 (13.9)	<0.0001	4044 (11.9)	<0.0001
Previous cancer, *n* (%)	24,336 (10.5)	5557 (8.0)	13,438 (10.5)	<0.0001	5341 (15.7)	<0.0001
CHA_2_DS_2_VASc score, mean ± SD	1.2 ± 1.2	0.3 ± 0.5	1.9 ± 1.1	<0.0001	0.6 ± 0.7	<0.0001
HASBLED score, mean ± SD	1.1 ± 1.2	0.3 ± 0.7	1.6 ± 1.2	<0.0001	1.0 ± 1.1	<0.0001
Charlson index, mean ± SD	2.3 ± 2.8	0.8 ± 1.8	3.0 ± 3.0	<0.0001	2.4 ± 2.7	<0.0001
Frailty index, mean ± SD	4.1 ± 6.0	1.9 ± 3.6	5.1 ± 6.6	<0.0001	4.9 ± 6.3	<0.0001
Frailty index, mean ± SD	4.1 ± 6.0	1.9 ± 3.6	5.1 ± 6.6	<0.0001	4.9 ± 6.3	<0.0001

AF = atrial fibrillation; BBB = bundle branch block; CABG = coronary artery bypass graft; COPD = chronic obstructive pulmonary disease; ICD = implantable cardioverter defibrillator; MI = myocardial infarction; PCI = percutaneous coronary intervention.

**Table 2 jcm-11-01981-t002:** Clinical outcomes during entire follow-up in AF patients.

	Lone AF ≤ 60 yo	AF ≤ 60 yo with Known Cardiac Disease	AF ≤ 60 yo with Extra-Cardiac Comorbidities	AF > 60 yo
All-cause death	3518 (2.3)	17,332 (5.7)	4697 (6.2)	592,190 (13.7)
Cardiovascular death	486 (0.3)	5095 (1.7)	616 (0.8)	180,989 (4.2)
Non-cardiovascular death	3032 (2.0)	12,237 (4.0)	4081 (5.4)	411,201 (9.5)
Ischemic stroke	1030 (0.7)	3866 (1.3)	653 (0.9)	106,118 (2.5)
Rehospitalization for HF	1349 (0.9)	15,400 (5.6)	1165 (1.6)	381,871 (9.9)

AF = atrial fibrillation, yo = years old, HF = heart failure.

**Table 3 jcm-11-01981-t003:** Hazard ratios (95% CI) associated with type of AF for the incident outcome of interest.

	Hazard Ratio	*p*-Value	Adjusted * Hazard Ratio	*p*-Value
All-cause death				
AF ≤ 60 yo with known cardiac disease (vs Lone AF)	2.38 (2.30–2.47)	<0.0001	2.13 (2.05–2.21)	<0.0001
AF ≤ 60 yo with extra-cardiac comorbidities (vs Lone AF)	2.57 (2.46–2.68)	<0.0001	2.46 (2.36–2.57)	<0.0001
AF ≤ 60 yo with extra-cardiac comorbidities (vs AF with known cardiac disease)	1.08 (1.04–1.11)	<0.0001	1.15 (1.12–1.19)	<0.0001
AF > 60 yo (vs Lone AF)	5.46 (5.29–5.65)	<0.0001	5.52 (5.34–5.70)	<0.0001
Cardiovascular death				
AF ≤ 60 yo with known cardiac disease (vs Lone AF)	5.08 (4.63–5.58)	<0.0001	4.80 (4.37–5.27)	<0.0001
AF ≤ 60 yo with extra-cardiac comorbidities (vs Lone AF)	2.44 (2.17–2.75)	<0.0001	2.39 (2.12–2.69)	<0.0001
AF ≤ 60 yo with extra-cardiac comorbidities (vs AF with known cardiac disease)	0.48 (0.44–0.52)	<0.0001	0.50 (0.46–0.54)	<0.0001
AF > 60 yo (vs Lone AF)	11.99 (10.97–13.11)	<0.0001	11.64 (10.65–12.73)	<0.0001
Non-cardiovascular death				
AF ≤ 60 yo with known cardiac disease (vs Lone AF)	1.95 (1.88–2.03)	<0.0001	1.72 (1.65–1.79)	<0.0001
AF ≤ 60 yo with extra-cardiac comorbidities (vs Lone AF)	2.59 (2.47–2.71)	<0.0001	2.47 (2.35–2.59)	<0.0001
AF ≤ 60 yo with extra-cardiac comorbidities (vs AF with known cardiac disease)	1.33 (1.28–1.37)	<0.0001	1.44 (1.39–1.49)	<0.0001
AF > 60 yo (vs Lone AF)	4.42 (4.26–4.58)	<0.0001	4.54 (4.38–4.70)	<0.0001
Ischemic stroke				
AF ≤ 60 yo with known cardiac disease (vs Lone AF)	1.85 (1.73–1.98)	<0.0001	1.72 (1.61–1.85)	<0.0001
AF ≤ 60 yo with extra-cardiac comorbidities (vs Lone AF)	1.24 (1.13–1.37)	<0.0001	1.21 (1.09–1.33)	<0.0001
AF ≤ 60 yo with extra-cardiac comorbidities (vs AF with known cardiac disease)	0.67 (0.62–0.73)	<0.0001	0.69 (0.64–0.75)	<0.0001
AF > 60 yo (vs Lone AF)	3.54 (3.33–3.76)	<0.0001	3.35 (3.15–3.56)	<0.0001
Rehospitalization for HF				
AF ≤ 60 yo with known cardiac disease (vs Lone AF)	5.97 (5.65–6.31)	<0.0001	5.76 (5.44–6.09)	<0.0001
AF ≤ 60 yo with extra-cardiac comorbidities (vs Lone AF)	1.69 (1.56–1.83)	<0.0001	1.67 (1.55–1.81)	<0.0001
AF ≤ 60 yo with extra-cardiac comorbidities (vs AF with known cardiac disease)	0.28 (0.27–0.30)	<0.0001	0.29 (0.27–0.31)	<0.0001
AF > 60 yo (vs Lone AF)	10.30 (9.77–10.87)	<0.0001	9.93 (9.41–10.47)	<0.0001

* Adjustment on age and sex. For comparison with age >60, adjustment is on sex only.

## Data Availability

The data and study materials will not be made available to other researchers for purposes of reproducing the results or replicating the procedure. Because this study used data from human subjects, the data and everything pertaining to the data are governed by the French Health Agencies and cannot be made available to other researchers.

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
