# Peer review of "Prognosis of Atrial Fibrillation with or without Comorbidities: Analysis of Younger Adults from a Nationwide Database"

_jcm, 2022, doi:10.3390/jcm11071981_

Round 1
Reviewer 1 Report
This is a simple, but robust study evaluating prognostic significance of various risk factors in AF. Its major strength is large number of patients: more than 2 million! Its major weakness is that all those relations are quite well known (see e.g. https://heart.bmj.com/content/early/2022/01/19/heartjnl-2021-320036). The results are presented clearly and the discussion is focused. I have some specific comments.
- Abstract: is messy. Coma is missing in 2203,702, why there is «Lone» AF” and “Lone AF”? What is HD?
- The division into groups is not clear from the text. Do groups 2-4 include only patients younger than 60 y? If so, it should be clearly stated in the Abstract and Methods. It is only evident from the figure.
- Figures 2 & 3: Figure titles should be above respective panels.
Author Response
Thanks for this comment. All grammatical corrections have been done and highlighted in the new manuscript

Reviewer 2 Report
Reviewer reply to” Prognosis of atrial fibrillation with or without comorbidities. Analysis of younger adults from a nation-wide database”.
In this manuscript, Mertz et al evaluated the prognosis of patients hospitalized with atrial fibrillation( AF ) with and without known cardiac disease( KCD ) or extra cardiac comorbidities ( ECC ), and to determine the prognostic value of these conditions associated with AF. The results of this study showed that patients in the ‘ Lone AF ‘ group remain severe in terms of cardiovascular(CV ) events but still with a lower incidence than the patient with associated KCD or ECC.
This study is well designed, the results are clearly described and are fully discussed.
Minor
1. Page 4, line 36 The incidence of CV death -> The incidence of non-CV death
2. Page 4, line 39
in the AF with KCC group-> in the AF with KCD group 3. Page 4, line 43 with AF-ECC or AF-ECC -> with AF-KCD or AF-ECC 4. Page 5, line 15 with AF-ECC and AF-KCC->with AF-ECC and AF-KCD 5. Page 7, line 24 with an incidence than the patient with-> with an incidence lower than the patient with
6. Page 10, line 2 know cardiac disease ->known cardiac disease 7. Page 11,line 2 know cardiac disease-> known cardiac disease 8. Page 12,line 2 know cardiac disease-> known cardiac disease 9. Page 12,line 17 KCD and ECCC-> KCD and ECC 10. Page 13,line 3 know cardiac disease-> known cardiac disease 11. Page 14,line 2 know cardiac disease->known cardiac disease 12. Page 17,line 4 know cardiac disease->known cardiac disease 13. Page 18, line 6, 11, 16, 21 extra-cardiac disease-> extra-cardiac comorbidities
Author Response

(The authors gave the same response as above.)
